# COVID-19 Infection May Drive EC-like Myofibroblasts towards Myofibroblasts to Contribute to Pulmonary Fibrosis

**DOI:** 10.3390/ijms241411500

**Published:** 2023-07-15

**Authors:** Xiuju Wu, Daoqin Zhang, Kristina I. Boström, Yucheng Yao

**Affiliations:** 1Division of Cardiology, David Geffen School of Medicine at UCLA, Los Angeles, CA 90095, USA; 2Department of Pediatrics, Stanford University, Stanford, CA 94305, USA; 3The Molecular Biology Institute at UCLA, Los Angeles, CA 90095, USA

**Keywords:** COVID-19, endothelial cells, pulmonary fibrosis

## Abstract

COVID-19 has an extensive impact on Homo sapiens globally. Patients with COVID-19 are at an increased risk of developing pulmonary fibrosis. A previous study identified that myofibroblasts could be derived from pulmonary endothelial lineage cells as an important cell source that contributes to pulmonary fibrosis. Here, we analyzed publicly available data and showed that COVID-19 infection drove endothelial lineage cells towards myofibroblasts in pulmonary fibrosis of patients with COVID-19. We also discovered a similar differentiation trajectory in mouse lungs after viral infection. The results suggest that COVID-19 infection leads to the development of pulmonary fibrosis partly through the activation of endothelial cell (EC)-like myofibroblasts.

## 1. Introduction

Since the start of the global pandemic, there have been over a hundred million confirmed cases of COVID-19 infection and almost seven million COVID-19-related deaths (https://www.who.int/publications/m/item/weekly-epidemiological-update-on-covid-19, (accessed on 5 May 2023). Although most patients with COVID-19 infection experience zero or mild symptoms, for others it has caused severe damages to their lungs [1,2]. Recent studies show a dramatic increase in COVID-19-related pulmonary fibrosis, and this poses a new threat to COVID-19 patients [3]. Pulmonary fibrosis is a severe fibrotic lung disease, in which extensive fibrogenesis occurs in the interstitium of lung tissues occupying the alveolar gas exchange units, leading to the reduction in pulmonary function [4,5]. Several types of cells secrete growth factors and cytokines that activate myofibroblasts, which produce aberrant compositions of the extracellular fibrotic matrix outlining the progress of pulmonary fibrosis [6]. Initially observed to contribute to tissue repair after injury, myofibroblasts are characterized to have the features of both fibroblasts and smooth muscle cells [7]. Beyond normal tissue repair, extensive amounts of myofibroblasts are found in fibrosis of the liver, kidneys, heart, and lungs where they persistently produce unwanted extracellular fibrotic matrix, including fibrillar collagens, fibronectin, and other fibrotic proteomes.

Pulmonary ECs are critical components of lung tissue. The absence of ECs causes undeveloped lungs and endothelial defects that disable lung repair after injury [8]. In a recent report, a novel differentiation trajectory was identified in normal lung tissue, where pulmonary ECs can differentiate into EC-like myofibroblasts towards myofibroblasts [9]. The study shows that EC-like myofibroblasts significantly contribute myofibroblasts to pulmonary fibrosis [9]. In this study, we hypothesize that this differentiation pathway contributes to COVID-19-related pulmonary fibrosis. To test this hypothesis, we analyzed the publicly available single-cell RNA sequencing (scRNA-seq) data of patients with severe pulmonary fibrosis after COVID-19 infection [10]. The results showed a robust elevation in EC-like myofibroblasts in these cases and identified COVID-19 infection as a force that drives ECs to EC-like myofibroblasts and myofibroblasts.

## 2. Results

### 2.1. Analysis of scRNA-Seq Reveals That COVID-19 Infection Drives ECs towards EC-like Myofibroblasts and Myofibroblasts and Contributes to Pulmonary Fibrosis of COVID-19 Patients

To examine the effect of COVID-19 infection on pulmonary endothelial lineage cells, we analyzed publicly available data, which contained one pulmonary scRNA-seq dataset from ten normal human lungs and three patients with COVID-19-related pulmonary fibrosis [9]. The Uniform Manifold Approximation and Projection (UMAP) revealed significant alterations in the composition of pulmonary endothelial lineage cells in these patients (Figure 1a). The percentage of EC-like myofibroblasts that expressed both endothelial and myofibroblast markers, ACTA2, CCN2, POSTN, COL1A1, COL3A1 FN1, and PDGFRα [11,12,13,14] (cluster 7), and myofibroblasts (cluster 8) were robustly increased in these patients (Figure 1a–c). The percentages of different EC types were altered. EC 2 (cluster 1) and capillary EC 2 (cluster 6) were increased. EC 1 (cluster 0) and EC 3 (cluster 2), capillary ECs 1 and 3 (clusters 3 and 5), and arterial ECs (cluster 4) were decreased (Figure 1a–c). Interestingly, the cell differentiation trajectory projected a clear direction that originated from capillary EC 1 (cluster 5) to capillary EC 2 (cluster 6), towards EC-like myofibroblasts (cluster 7), and ending with myofibroblasts (cluster 8) (Figure 1d). COVID-19 infection elevated capillary EC 2 about two-fold to promote the transition from ECs to myofibroblasts (Figure 1c). Following this trajectory direction, gene expression dynamics showed a decrease in endothelial markers with an increase in myofibroblast markers (Figure 1e and Appendix A). The upregulated CCN2 and POSTN in EC 1 and EC 2 suggested that COVID-19 infection induced an excess production of extracellular matrix in ECs and contributed to fibrosis [15,16]; however, the differentiation of EC 1 and EC 2 were not connected to myofibroblasts (Appendix A). Previous studies have shown that COVID-19 could trigger a cytokine storm in lung tissues [17,18,19,20,21]. We analyzed the expressions of cytokines, such as IL1β, IL6, IL11, and IL33. We found that expressions of IL1β, IL6, and IL11 were very low in ECs and EC-derived myofibroblasts, and there was no change in IL33 expression due to COVID-19 infection in these cell clusters (Figure 1b). This is consistent with previous reports where other cell types are responsible for cytokine secretion in COVID-19 infected lungs, such as immune cells [17,18,19,20,21]. Together, the results suggested that COVID-19 caused a cell transition from ECs towards EC-like myofibroblasts and myofibroblasts to contribute to pulmonary fibrosis in COVID-19 patients. The analysis also identified an induction of MGP in EC 3, capillary ECs 1 and 2, EC-like myofibroblasts, and myofibroblasts (Figure 1b). A previous study suggested that MGP regulates the activity of Bone Morphogenic Protein-1 (BMP-1) and controls the differentiation of EC-like myofibroblasts [9]. Here, the induction of MGP in ECs, EC-like myofibroblasts, and myofibroblasts indicated that the activation of this differentiation pathway by COVID-19 infection might induce MGP to regulate this unwanted activation.

### 2.2. Analysis of scRNA-Seq Uncovers That Influenza A Viral Infection Drives ECs towards EC-like Myofibroblasts

To determine if other viral infections affected pulmonary endothelial lineage cells, we analyzed another set of publicly available scRNA-seq data, which were obtained from mouse lungs with the influenza A viral infection [22]. We analyzed the endothelial lineage cells. The UMAP identified five EC clusters (clusters 0, 1, 3, 4, and 6) and three clusters of EC-like myofibroblasts 1, 2, and 3 (clusters 2, 5, and 7) (Figure 2a–c). The composition of EC-like myofibroblasts (clusters 2, 5, and 7) was dramatically increased after one day of viral infection and gradually increased as the infection continued (Figure 2d). EC-like myofibroblasts 1, 2, and 3 expressed endothelial markers and different levels of myofibroblast markers, suggesting that these cells were undergoing the transition to myofibroblasts (Figure 2b,c and Appendix A). EC-like myofibroblast 1 (cluster 5) expressed Acta2, Tagln, Postn, Col1a1, and Col3a1 but showed no expression of Pdgfra and Fibronectin 1 (Fn1), suggesting these cells were in early transition towards myofibroblasts. EC-like myofibroblast 2 (cluster 7) expressed less Acta2, Tagln, Col1a1, and Col3a1 but expressed more Pdgfra and Fn1, suggesting these cells were in mid-transition towards myofibroblasts. EC-like myofibroblast 3 (cluster 2) expressed more Pdgfra, Fn1, Col1a1, and Col3a1 but had no expression of Acta2 and Tagln, suggesting that these cells were in a late transition to myofibroblast (Figure 2c). Compared to a mock infection, the cell differentiation trajectory after a 3-day infection confirmed the pathway from EC-like myofibroblasts 1 to 2, ending with 3 (Figure 2e). The percentages of ECs were again altered. The percentages of ECs 1, 2, and 4 (clusters 0, 1 and 4) were decreased and ECs 3 and 5 (clusters 3 and 6) were increased (Figure 2d), suggesting that specific EC populations contributed to the EC-like myofibroblasts. The cell differentiation trajectory along the time course of viral infection suggested that EC-like myofibroblasts 1, 2, and 3 were derived from EC 2 (Figure 2e,f). The analysis also showed a specific pattern of MGP expression in EC 2 and EC-like myofibroblasts 1, 2, and 3 along the time course of infection (Figure 2b), again supporting that MGP may regulate the transition from ECs to EC-like myofibroblasts.

## 3. Discussion

While we have survived the worst of the COVID-19 pandemic, COVID-19 still poses a significant threat to people all over the world by causing injuries to multiple organs. Severe pulmonary fibrosis after COVID-19 infection has become a critical issue for the late-stage and long COVID-19 patients. Unfortunately, in some cases, lung transplantation may be the only clinical option [9]. Previous studies suggested that the inflammatory storm triggered by a COVID-19 infection activates an excess of myofibroblasts resulting in pulmonary fibrosis [23]. Here, we showed a clear increase in EC-like myofibroblasts, which transition towards myofibroblasts in pulmonary fibrosis ofCOVID-19 patients. We previously identified that excess TGF-beta drives EC-like myofibroblasts towards myofibroblasts, thereby contributing to fibrosis [9]. We argue that over-expressed pulmonary TGF-beta after COVID-19 infection could be the key driver in this case. COVID-19 patients with pulmonary fibrosis show an accumulation of *KRT17^+^* epithelial cells, which express high levels of TGF-beta [9]. Immune cells responding to COVID-19 infection are also reported to produce a large amount of TGF-beta [24]. In addition, other pulmonary cells injured by COVID-19 may secrete more TGF-beta aimed for repair [24]. Excessive TGF-beta produced from all these cellular sources is able to alter the micro-environment and thus drive ECs and EC-like myofibroblasts towards unwanted myofibroblast fates. In this study, similar differentiation trajectories of ECs and EC-like myofibroblasts were found in mouse lungs after influenza A virus infection. The results suggest that the contribution of ECs and EC-like myofibroblasts to fibrosis after a variety of viral infections may have similar mechanisms.

We have shown that the lack of MGP caused an aggressive differentiation of EC-like myofibroblasts towards myofibroblasts [9]. We discovered that MGP binds to BMP-1 and inhibits its activity in activation of TGF-beta [9]. In this analysis, we identified the induction of MGP in the differentiation trajectory from ECs to EC-like myofibroblasts and myofibroblasts. It would be interesting to further investigate the alteration in activity of pulmonary BMP-1 after COVID-19 infection. Berbamine has been reported to prevent EC-like myofibroblasts from transitioning into myofibroblasts and reducing the pulmonary fibrosis in animal models [9]. Berbamine is a compound extracted from plants and can be purchased as a supplement. It would be interesting to study the effect of berbamine on the pulmonary fibrosis of COVID-19 patients.

## 4. Methods

### Single Cell RNA-Seq Analysis

The EC lineages derived from ten controls, and three COVID-19 patients were used for further analysis. CD45-CD31+ and CD45-VE-cadherin+ cells of whole mouse lungs were analyzed 1-, 3-, and 6-days post influenza A virus infection and mock infection. Scanpy v1.9.3 was used to process human data, and R package Seurat (v4.3.0) was used to process mouse data. Human data were trained using a deep learning neural network model, scvi.model.SCVI, from single-cell Variational Inference (scVI) tools. Dimension reduction using UMAP, cell clustering, and data visualization were then performed. A subset of data was further analyzed to construct single-cell trajectory using R package Monocle3 (v1.3.1). The cells were ordered along a learned trajectory, and the expression dynamics of interested genes were plotted along the pseudotime.

## Figures and Tables

**Figure 1 ijms-24-11500-f001:**
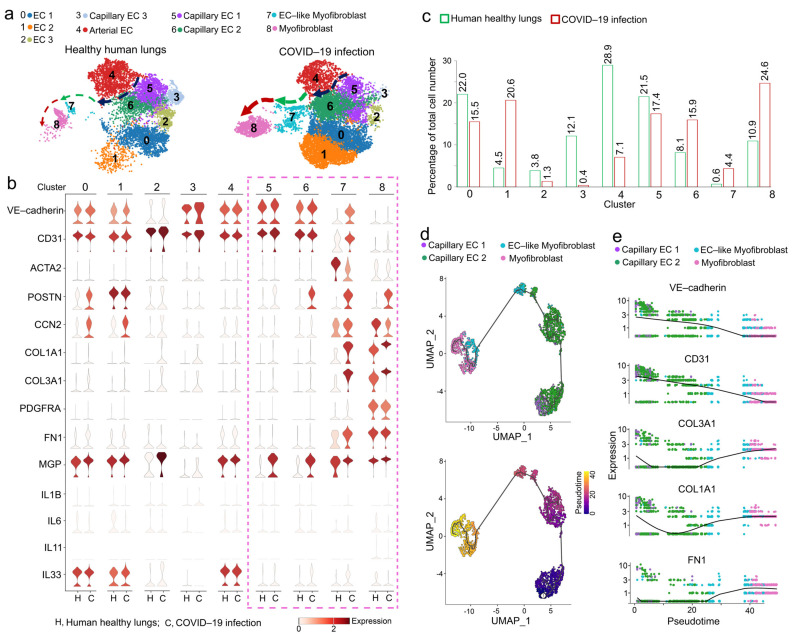
ScRNA-seq identifies excess EC-like myofibroblasts differentiating towards myofibroblasts in pulmonary fibrosis of COVID-19 patients. (**a**) UMAP for the cell populations subclustered from the whole population of pulmonary cells. Arrows indicate the differentiation trajectory with alterations in cell compositions. (**b**) Violin plots of gene expression of lineage markers. Fn1, fibronectin 1. (**c**) Cell compositions of different populations in lungs of healthy humans or COVID-19 patients. (**d**) Pseudotemporal trajectories of the cell clusters. (**e**) The expression of genes along the single cell trajectories.

**Figure 2 ijms-24-11500-f002:**
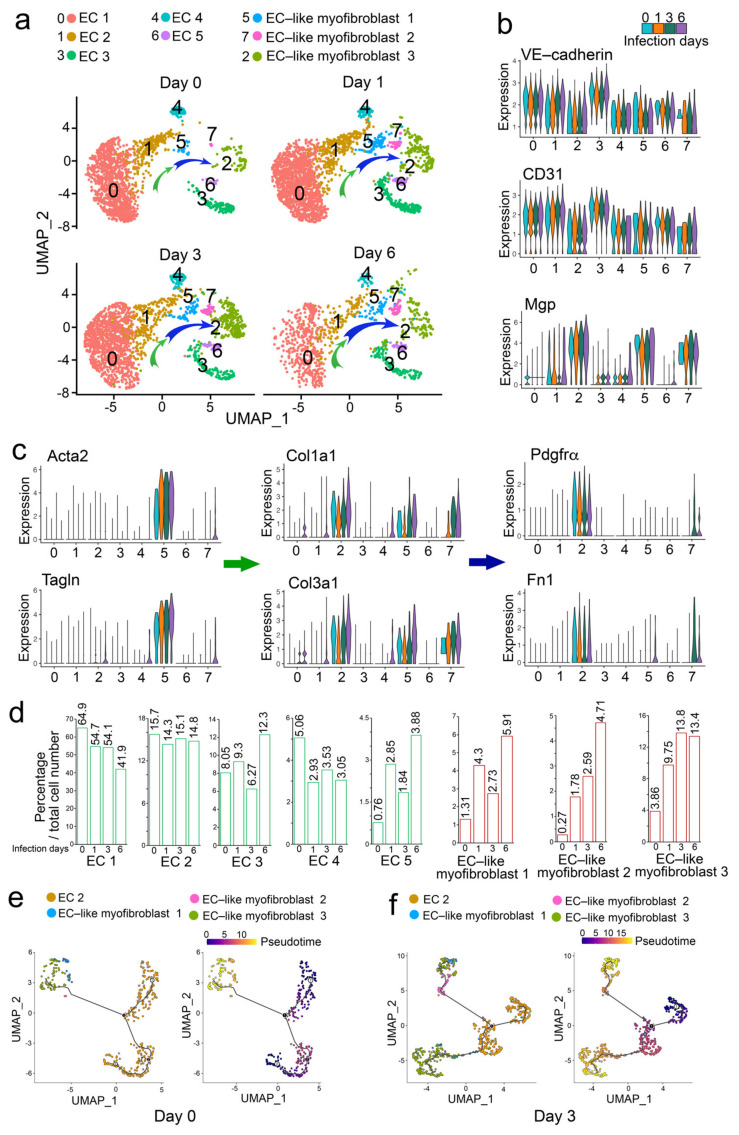
ScRNA-seq reveals ECs differentiating towards myofibroblasts in mouse lungs after influenza A viral infection. *(***a**) UMAP for the cell populations subclustered from CD45-pulmonary cells along the time course of influenza A viral infection. Arrows indicate the differentiation trajectory with alterations in cell compositions. (**b**) Violin plots of the gene expression of VE-cadherin, CD31 and MGP in subclusters. (**c**) Violin plots of the gene expression of the myofibroblast lineage markers. Arrows indicate the markers of different stages of myofibroblast differentiation. *(***d**) Cell compositions of different EC lineage populations in mouse lungs along the time course of influenza A viral infection. (**e**,**f**) Pseudotemporal trajectories of the cell clusters at day 0 (mock infection) and 3 after influenza A viral infection.

## Data Availability

ScRNA-seq data of human lungs from patients with late stage COVID-19 and controls were obtained from the Gene Expression Omnibus (GEO) database with accession number GSE158127 (https://www.ncbi.nlm.nih.gov/geo/query/acc.cgi?acc=GSE158127, accessed on 5 May 3023). The mouse data were downloaded from the NCBI BioProject database with accession number PRJNA612345 (https://www.ncbi.nlm.nih.gov/sra?linkname=bioproect_sra_all&from_uid=612345, accessed on 5 May 3023).

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
