# Peer review of "COVID-19 Infection May Drive EC-like Myofibroblasts towards Myofibroblasts to Contribute to Pulmonary Fibrosis"

_ijms, 2023, doi:10.3390/ijms241411500_

Round 1

Reviewer 1 Report

This study aimed to analyze publicly available data, with the ultimate goal of testing whether COVID-19 infection could drive ECs towards EC-like myofibroblasts and myofibroblasts to contribute to pulmonary fibrosis of COVID-19 patients

 The authors reported that COVID-19 infection drove endothelial-lineage cells towards myofibroblasts in pulmonary fibrosis of patients with COVID-19. The authors concluded that COVID-19 infection leads to the development of pulmonary fibrosis partly through the activation of endothelial cell (EC)-like myofibroblasts.

This is an interesting study. Overall, it is a clear, and concise, and manuscript. The introduction is relevant. Sufficient information about the previous study findings is presented for readers to follow the present study rationale and procedures.  The results are clear.  The information presented represents valuable information.

None

Author Response

Reviewer 1

“This study aimed to analyze publicly available data, with the ultimate goal of testing whether COVID-19 infection could drive ECs towards EC-like myofibroblasts and myofibroblasts to contribute to pulmonary fibrosis of COVID-19 patients. The authors reported that COVID-19 infection drove endothelial-lineage cells towards myofibroblasts in pulmonary fibrosis of patients with COVID-19. The authors concluded that COVID-19 infection leads to the development of pulmonary fibrosis partly through the activation of endothelial cell (EC)-like myofibroblasts. This is an interesting study. Overall, it is a clear, and concise, and manuscript. The introduction is relevant. Sufficient information about the previous study findings is presented for readers to follow the present study rationale and procedures.  The results are clear.  The information presented represents valuable information.”

Thank you very much for the comments.

Reviewer 2 Report

Overall, this appears to be a useful contribution.

My questions are:

1. The markers of "myofibroblasts" include: aSMA (ACTA2), CCN2 (CTGF), POSTN. The authors have chosen to focus on "synthetic fibroblast" markers such as PDGFRa and COL1A1. Without actually examining "myofibroblast" markers, the authors need to take care regarding describing the populations they have identified, which appear to be more likely to be "fibroblasts". In that regard, the authors should examine (for example, articles such as PMID: 32755548 PMID: 33981032. PMID: 24336287) fibroblast markers vs myofibroblast markers to better define the cell populations they are examining

2. This general idea has been proposed previously, based on cytokine profiles (PMID: 32406037 PMID: 32171076), specifically the overexpression of IL-6. I wonder if there is a possibility of examining: IL-1b, IL-6, IL-11, IL-33 etc?

Author Response

Reviewer 2

  1. “The markers of "myofibroblasts" include: aSMA (ACTA2), CCN2 (CTGF), POSTN. The authors have chosen to focus on "synthetic fibroblast" markers such as PDGFRa and COL1A1. Without actually examining "myofibroblast" markers, the authors need to take care regarding describing the populations they have identified, which appear to be more likely to be "fibroblasts". In that regard, the authors should examine (for example, articles such as PMID: 32755548 PMID: 33981032. PMID: 24336287) fibroblast markers vs myofibroblast markers to better define the cell populations they are examining”

We have analyzed more of the markers that were suggested by reviewer, including CTGF and POSTN. The results showed a similar expression pattern as for the other markers (Figure 1b).

  1. “This general idea has been proposed previously, based on cytokine profiles (PMID: 32406037 PMID: 32171076), specifically the overexpression of IL-6. I wonder if there is a possibility of examining: IL-1b, IL-6, IL-11, IL-33 etc?”

We have analyzed the expression of IL-1b, IL-6, IL-11, IL-33. The results showed no changes in the expression of these cytokines (Figure 1b).

Round 2

Reviewer 2 Report

The new information is interesting, but is inadequately explained and raises issues regarding interpretation. The new data needs to be appropriately referenced.

The paper relies on the identification of a "myofibroblast" and an "EC myofibroblast" population, yet the authors do not explain how (which markers) they used to identify these populations, and the bases for how these markers were chosen.

The usual myofibroblast marker is ACTA2, but this is not shown. Thus it is difficult to call these populations "myofibroblasts". The authors, based on previous suggestions, show POSTN and CCN2 (the authors use the archaic name "CTGF". This needs to be changed to "CCN2") suggesting perhaps a "protomyofibroblast" subtype.  They provide no references indicating why they have chosen these as markers. Moreover, no analysis or interpretation is provided regarding the upregulation of POSTN and CCN2 in EC1 and EC2 clusters. In any event, without using ACTA2  it is difficult to identify any of these cell populations as myofibroblast.  Similarly, the trajectory analysis in Figure 1D,E would be more convincing if myofibroblast markers such as CCN2 and POSTN are used. What would happen if the authors were to conduct something similar to Fig 1 D,E, but use EC1 and EC2 instead of Capillary EC1 and EC2.

The issue with ACTA2 is a bit odd, as the authors use ACTA2 in Figure 2. The authors need to show ACTA2 in Figure1, and it would be helpful to show at least POSTN in Figure 2

The authors do not present the current paper in the context of prior papers showing a cytokine storm in COVID, including the overexpression of IL-6, that would be anticipated to result in fibrosis. It is plausible that the methods used in the experiments analyzed here are not sensitive enough to detect the cells in the lung (immune cells) that are presumably the source of IL-6. Clearly, the EC and 'fibroblast' populations are not producing IL-6 (IL-6 is induced by TGFbeta in fibroblasts). The current data are not interpreted in the context of prior data,

Author Response

Reviewer 2:

The new information is interesting, but is inadequately explained and raises issues regarding interpretation. The new data needs to be appropriately referenced.

The paper relies on the identification of a "myofibroblast" and an "EC myofibroblast" population, yet the authors do not explain how (which markers) they used to identify these populations, and the bases for how these markers were chosen.

  1. “The usual myofibroblast marker is ACTA2, but this is not shown. Thus it is difficult to call these populations "myofibroblasts". The authors, based on previous suggestions, show POSTN and CCN2 (the authors use the archaic name "CTGF". This needs to be changed to "CCN2") suggesting perhaps a "protomyofibroblast" subtype. They provide no references indicating why they have chosen these as markers. Moreover, no analysis or interpretation is provided regarding the upregulation of POSTN and CCN2 in EC1 and EC2 clusters. In any event, without using ACTA2 it is difficult to identify any of these cell populations as myofibroblast.  Similarly, the trajectory analysis in Figure 1D,E would be more convincing if myofibroblast markers such as CCN2 and POSTN are used. What would happen if the authors were to conduct something similar to Fig 1 D,E, but use EC1 and EC2 instead of Capillary EC1 and EC2.”

ACTA2 expression has been added to Figure 1b, which shows its expression in EC-like myofibroblasts. We have changed the gene name “CTGF” to “CCN2”. We have added references regarding the markers that define myofibroblasts [11-14].

We have included in the text that “The upregulated CCN2 and POSTN in EC 1 and EC 2 suggested that COVID-19 infection induced excess production of extracellular matrix in ECs and contributed to fibrosis.”.  

We have added the expression of CCN2 and POSTN along the single cell trajectories to Supplemental Figure 1.

We used EC 1, EC 2, EC-like myofibroblasts, and myofibroblasts to construct trajectories. We found no projected connection between EC 1 or EC 2 and EC-like myofibroblasts or myofibroblasts (Supplemental Figure 2). We have added the text that “However, the differentiation of EC 1 and EC 2 were not connected to myofibroblasts (Supplemental Figure 2).”.

  1. “The issue with ACTA2 is a bit odd, as the authors use ACTA2 in Figure 2. The authors need to show ACTA2 in Figure1, and it would be helpful to show at least POSTN in Figure 2”

ACTA2 expression has been added to Figure 1. We also examined Postn expression in the mouse data and found the expression of Postn in EC-like myofibroblasts (Supplemental Figure 3).

  1. “The authors do not present the current paper in the context of prior papers showing a cytokine storm in COVID, including the overexpression of IL-6, that would be anticipated to result in fibrosis. It is plausible that the methods used in the experiments analyzed here are not sensitive enough to detect the cells in the lung (immune cells) that are presumably the source of IL-6. Clearly, the EC and 'fibroblast' populations are not producing IL-6 (IL-6 is induced by TGFbeta in fibroblasts). The current data are not interpreted in the context of prior data,”

We added the following statement to the results: “Previous studies have shown that COVID-19 could trigger a cytokine storm in lung tissue [17-21]. We analyzed the expression of cytokines, such as IL1b, IL6, IL11 and IL33. We found that expression of IL1b, IL6, and IL11 were very low in ECs and EC-derived myofibroblasts, and there was no change in IL33 expression due to COVID-19 infection in these cell clusters (Figure 1b). This would be consistent with previous reports that other cell types being responsible for cytokine secretion in COVID-19 infected lungs, such as immune cells [17-21].”
